# Perfect Optical Absorbers by All-Dielectric Photonic Crystal/Metal Heterostructures Due to Optical Tamm State

**DOI:** 10.3390/nano11123447

**Published:** 2021-12-20

**Authors:** Guang Lu, Kaiyuan Zhang, Yunpeng Zhao, Lei Zhang, Ziqian Shang, Haiyang Zhou, Chao Diao, Xiachen Zhou

**Affiliations:** 1School of Space Science and Physics, Shandong University, Weihai 264209, China; 17862701682@163.com (K.Z.); 15588318820@163.com (Y.Z.); 201920773@mail.sdu.edu.cn (L.Z.); 201920775@mail.sdu.edu.cn (Z.S.); Zhouhaiyang@163.com (H.Z.); physicsdc@163.com (C.D.); xcsduwh@163.com (X.Z.); 2Laboratory for ElectromAgnetic Detection (LEAD), Institute of Space Sciences, Shandong University, Weihai 264209, China

**Keywords:** optical Tamm state, perfect absorption, photonic crystals, heterostructure, metal film, interface

## Abstract

In this study, we theoretically and experimentally investigated the perfect optical absorptance of a photonic heterostructure composed of a truncated all-dielectric photonic crystal (PC) and a thick metal film in the visible regions. The three simulated structures could achieve narrow-band perfect optical absorption at wavelengths of 500 nm, 600 nm, and 700 nm, respectively. Based on the measured experimental results, the three experimental structures achieved over 90% absorption at wavelengths of 489 nm, 604 nm, and 675 nm, respectively. The experimental results agreed well with the theoretical values. According to electromagnetic field intensity distributions at the absorption wavelengths, the physical mechanism of perfect absorption was derived from the optical Tamm state (OTS). The structure was simple, and the absorption characteristics were not significantly affected by the thickness of the thick metal layer, which creates convenience in the preparation of the structure. In general, the proposed perfect absorbers have exciting prospects in solar energy, optical sensor technology, and other related fields.

## 1. Introduction

Optical absorbers have attracted considerable research attention for their potential to be used in applications, including solar energy [1,2], sensing [3,4], thermal radiation [5,6], imaging [7], and optical switching [8]. With the development of electromagnetic simulation and micro-nano processing technology, scientists have proposed a variety of perfect optical absorbers that can perfectly absorb light waves based on various micro/nano-structures, such as metamaterials [3,9,10], metasurface [11,12,13], and photonic crystals [14,15,16]. Micro/nano-structures that can excite the interface mode have also been used in the development of perfect optical absorbers in recent years. If an interface mode is excited in, or near, the lossy materials, the electric field distribution in the material can be greatly enhanced, improving absorption. The common interface modes are the defect mode [17,18] and surface plasmon polarization (SPP) [19,20,21]. In recent years, a new interface mode called the optical Tamm state (OTS) has provided a new method to achieve perfect absorption.

The OTS is a new kind of surface wave, confined at the interface between two different media, and it can also be called Tamm plasmon polarization (TPP) [22], It is derived from an analogy of the electronic Tamm state in solid-state physics [23]. Compared with traditional surface waves, such as surface plasmon polariton (SPP) [24,25], OTS can be excited directly by both TE- and TM-polarized waves and a specific incident angle is not required [22,26,27,28]. These characteristics improve the application prospects of OTS, such that it has become a popular research topic in recent years. A large number of OTS-based absorption devices have been proposed. Metal-photonic crystal (M-PC) structures [27,28] and heterostructures composed of two photonic crystals (PC-PC) [22] are two ideal structures for exciting OTS. The research on OTS absorbers has mostly been focused on these two structures. Generally, thin metal or thin lossy material layers are placed on a photonic crystal [29,30,31,32] or are inserted at the interface [15,33,34]. The absorption characteristics of these structures are very sensitive to the thickness of the thin metal or the thin lossy material layer. A complex process is required to produce lossy-material nano layers with good optical properties in a laboratory setting. Scaling the process up to large-scale production will bring added difficulties.

We propose a method for producing an all-dielectric PC on a thick layer of Ag and achieve perfect narrow-band absorption through the excitation of OTS. We designed the experimental samples and measured their properties. The experimental results were in good agreement with the theoretical results. When the thickness of the metal layer exceeded a certain level, the structure’s perfect absorption was not affected, which is more convenient for sample preparation and future applications. In Section 2, we analyze the conditions for OTS excitation and introduce the structural parameters for a perfect absorber. In Section 3, we present the simulated and experimental results. Finally, conclusions are given in Section 4.

## 2. Theory and Structure

In order to excite the OTS in a composite structure composed of two highly reflective materials, the phase-matching condition must be satisfied [27]. When light is transmitted into the composite structure, a portion of the light is reflected and the interface of the two highly reflective materials becomes a microcavity structure, as shown in Figure 1. The reflection coefficients of the material on the left and right sides of the microcavity are $r_{L}e^{{i\Phi }_{L}}$ and $r_{R}e^{{i\Phi }_{R}}$, respectively. When the optical Tamm state is formed, it can be considered that the light was reflected back and forth in the microcavity, the multiple reflected waves can be coherently enhanced so as to realize the resonance enhancement of light waves in the cavity. Due to the discontinuity of the refractive index between the left material and the right material, a phase Φ change will occur when light is transmitted in the microcavity. The specific relationship is as follows:(1)$$A\left[\begin{array}{c} 1\\ r_{L}e^{{i\Phi }_{L}} \end{array}\right]=\left[\begin{array}{cc} e^{i\Phi } & 0\\ 0 & e^{-i\Phi } \end{array}\right]\left[\begin{array}{c} r_{R}e^{{i\Phi }_{R}}\\ 1 \end{array}\right]$$
where A is the elimination constant. Formula (1) can be further simplified as:(2)$$r_{L}r_{R}e^{{i(\Phi }_{L}{+\Phi }_{R})}e^{i(2\Phi )}=1$$

When the two structures are closely connected, the length of the air cavity is 0 and the phase change is 0, thus resulting in Formula (3):(3)$${\;r}_{L}r_{R}e^{{i(\Phi }_{L}{+\Phi }_{R})}=1$$

The reflection amplitude of a highly reflective material is approximately 1, hence Formula (3) can be simplified as:(4)$${\;\Phi }_{L}{+\Phi }_{R\;}=0$$

According to Formula (4), OTS can be excited in a composite structure.

In our design, the photonic heterostructures of the absorbers were denoted by the structure (BA)^N^MS, as shown in Figure 2. (BA)^N^ represents the truncated dielectric PC, where A and B represent two different dielectrics and N represents the periodic number. A and B represent the TiO_2_ and SiO_2_ layers with refractive indices of n_A_ and n_B_ and thicknesses of d_A_ and d_B_, respectively. The refractive indices of the TiO_2_ and SiO_2_ layers were calculated based on the measured transmission spectra of the monolayer films of these materials, and values of n_A_ = 2.123 and n_B_ = 1.431 were obtained, respectively [32]. M represents the thick metal Ag layer with a thickness of d_M_. 

A Lorentz–Drude model [32] was used to describe the permittivity of the dispersive Ag as follows:(5)$$\varepsilon_{M}{=1-f}_{0}\frac{\omega_{P}^{2}}{\omega^{2}{+i\gamma }_{0}\omega }{-f}_{1}\frac{\omega_{P}^{2}}{\omega^{2}-\omega_{1}^{2}{+i\gamma }_{1}\omega }$$
where *f*_0_ = 0.9126, *ω_p_* = 1.3666 × 10^16^ Hz, *γ*_0_ = 1.140 × 10^16^ Hz, *f*_1_ = 0.3718, *ω*_1_ = 0.5275*ω_p_* and *γ*_1_ = 1.2920 × 10^15^ Hz. All parameter values were extracted from experimental measurements of a single silver layer. S represents the substrate, which is BK7 glass and has a refractive index of n_S_ = 1.52. The transmission (reflection) spectra were simulated using the transfer matrix method [35,36,37]. The absorptance can be calculated using the formula A = 1 − T − R, where R, T, and A are the reflectance, transmittance, and absorptance, respectively.

## 3. Results and Discussion

### 3.1. Simulated Results

To realize a perfect absorption structure, denoted by (BA)^N^MS at a wavelength of 600 nm, we selected the number of periods to be N = 8. The thicknesses of A, B, and M were chosen to be d_A_ = 63.2 nm, d_B_ = 93.8 nm, and d_M_ = 200 nm. The interface of the heterogeneous structure divided the structure into a section containing PC (AB)^8^ and another section containing the Ag-glass structure, MS. The reflectance and reflection phase of (AB)^8^ and MS were calculated. Since OTS was excited at the interface of the two structures, only the incident light from layer A could be considered when calculating the reflection phase of PC because layer A is connected to the metal layer.

From the reflection spectra presented in Figure 3a, it can be seen that the PC and metal layer M were highly reflective at a wavelength of 600 nm. The calculated reflection phase is shown in Figure 3b, where the black line is the reflection phase of (AB)^8^, Φ_(BA)_^8^, the red line is the reflection phase of the 200-nm Ag layer, Φ_MS_, and the blue line is the sum of the two phases, Φ_(BA)_^8^ + Φ_MS_. At a wavelength of 600 nm, the sum of the reflection phases is Φ_(BA)_^8^ + Φ_MS_ = 0, which meets the phase-matching condition of the excited OTS.

The truncated photonic crystal and metal layer were combined into a heterostructure and the reflectance (R) and absorptance (A) spectra of the structure were simulated. Figure 4a shows the simulated spectra for structure (BA)^8^MS. The red solid line represents the simulated absorptance, and maximum absorption A = 99.9% was achieved at a wavelength of 600 nm. The black solid line represents the simulated reflectance, and it can be seen that the minimum reflection was R = 0.1%. It can be stated that the structure can perfectly absorb light waves that have a wavelength of 600 nm.

To determine the physical mechanism of perfect absorption in the structure, the intensity distributions of the EM fields were simulated for an absorption wavelength of 600 nm, and the results are shown in Figure 4b. The intensity of the incident electric field was assumed to be 1. The red and blue solid lines indicate the intensity distributions of |E|^2^ and |H|^2^, respectively. The maximum values of the EM field intensities were localized at the interface between the truncated symmetric PC and the thick metal layer, which indicated that these types of localized modes originated in the OTS. It was thanks to this strong localized electric field in the interface and near the metallic layers that perfect absorption was achieved.

In order to analyze the influence of parameter errors on the perfect absorption characteristics of the structure, the effects of different structural parameters on the absorption characteristics were simulated. Firstly, the thicknesses of dielectric layers B and A were kept constant, while the thickness of metal layer M was varied, thereby simulating the absorption spectrum of the structure. Figure 5a shows the absorption spectrum of the metal layer for thicknesses of 20 nm, 40 nm, 60 nm, 80 nm, 100 nm, 150 nm, 200 nm, and 300 nm. The inset in Figure 5a shows the absorptance of the absorption peak versus the thickness of metal layer M. It can be seen that, when the thickness of the metal layer was less than 150 nm, the absorption rate of the structure at the resonance wavelength increased with the increase in thickness of the M layer; however, after the thickness of the M layer exceeded 150 nm, the absorption rate was consistently greater than 99%. Therefore, when the thickness of metal layer M exceeded a certain value, the perfect absorption characteristics of the structure were not affected by the thickness of the metal layer. Figure 5b shows the absorption spectrum of the structure for different thicknesses of the dielectric layers. For the simulations, when the thickness of layer B was kept unchanged, the black and green line showed the absorption spectrum of the structure for when the thickness of layer A was increased by 10 nm and decreased by 10 nm, respectively. When the thickness of layer A was kept unchanged, the red line and purple line show the absorption spectra of the structure for when the thickness of layer B was increased by 10 nm and decreased by 10 nm, respectively. The blue line and yellow line are the absorption spectra when the thickness of layers A and B were both increased by 10 nm and decreased by 10 nm. These characteristics of the structure allow for a certain level of convenience when preparing experimental samples; we only need to ensure that the thickness of the metal layer is slightly thicker than 150 nm.

The influence of a different period number of N on the absorption of the structure of (BA)^8^MS is shown in Figure 6a. The thickness of each layer is the same as that in Figure 4. It can be seen that when the period number is N = 8, the structure has the highest absorption. When N > 8, with the increase in N, the maximum absorption decreases gradually, and the absorption peak moves to a shorter wavelength. When N < 8, with the decrease in N, the maximum absorption rate decreases gradually, and the absorption peak moves to a longer wavelength. We simulated absorption spectra of (BA)^8^MS using different dielectric layer thicknesses, which are shown in Figure 6b. When the thickness of all dielectric layers increased by 1%, 2%, 3%, 4% and decreased by 1%, 2%, 3%, 4%, the absorption peak shifted to 606 nm, 612 nm, 617 nm, 623 nm, 594 nm, 588 nm, 583 nm, and 577 nm, respectively, and the offset of the absorption peak was +1%, +2%, +2.8%, +3.8%, −1%, −2%, −2.8%, −3.8%, respectively. It can be seen that the change in the dielectric layer thickness was approximately equal to the offset of the absorption peak. When the thickness of the dielectric layer was shifted by a small amount, the structure could also achieve near perfect absorption.

A perfect light absorber was designed for a wavelength of 700 nm by optimizing the structural parameters. The number of periods was selected as N = 7 and the structure was denoted as (BA)^7^MS. The thicknesses of A and B were set to d_A_ = 110.7 nm and dB = 74.6 nm, respectively, while the thickness of M remained constant. In the simulated spectra, the maximum absorption, A = 99.1%, and the minimum reflection, R = 0.9%, were observed at a wavelength of 700 nm and are shown in Figure 7a. The intensity distributions of the EM fields were simulated for an absorption wavelength 700 nm and the results are shown in Figure 7b. The maximum values of the EM field intensities were localized in the interface between the truncated symmetric PC and the thick metal layer. This OTS led to a perfect absorption at a wavelength of 700 nm. We reduced the thicknesses of A and B to d_A_ = 77.4 nm and d_B_ = 52.2 nm, respectively, and the number of N was changed to 10. We simulated reflection and absorption spectra of (BA)^10^MS, as shown in Figure 8a. The maximum absorption, A = 99.4%, and the minimum reflection, R = 0.6%. The structure could nearly perfectly absorb light at a wavelength of 500 nm. The simulated electromagnetic distribution is shown in Figure 8b. The maximum values of the EM field intensities were localized in the interface due to OTS.

### 3.2. Experimental Results

Three heterostructures were fabricated based on the designed parameters. The preparation of the structures was divided into two parts. First, a 200-nm thick silver layer was prepared on a BK7 substrate in a high vacuum environment, using a Rankuum Machinery ZZS900-3/G Box-type Vacuum Coating Machine. When the vacuum degree of the cavity reached 7.0 × 10^−4^ Pa, evaporation began. The temperature was set at 150 °C. The deposition rate of Ag was 46 nm/min. There was no assisting gas added in the vacuum cavity. We opened the vacuum chamber and placed the substrate BK7 near the silver mirror. Then, high vacuum conditions were re-established, and the PCs were prepared. The alternating A and B layers of the truncated PCs, with a (BA)^N^ structure were deposited on the planar face of the substrates (MS and S) using ion-assisted electron-beam evaporation under a high vacuum. The ion source was a Ø12 cm Kaufman ion source. During deposition, the vacuum chamber was kept at 2.0 × 10^−2^ Pa, the temperature was set at 150 °C, Ar + O_2_ ion assistance with an optimized oxygen partial pressure (1.0 × 10^−2^ Pa) was applied, the gas flow rates of Ar and O_2_ were 6.0 SCCM and 7.0 SCCM, respectively. An ion kinetic energy of 70 eV was kept for all layers. The deposition rates of TiO_2_ and SiO_2_ were 31 nm/min and 55 nm/min, respectively. The thicknesses of the M, A and B layers were monitored using quartz crystal sensors. The heterostructure (BA)^N^MS and contrast structure (BA)^N^S were prepared simultaneously.

The reflectance spectra of the (BA)^N^MS and (BA)^N^S samples were measured using a Perkin Elmer LAMBDA 1050 ultraviolet-visible-near-infrared spectrophotometer, and the results are shown in Figure 9. The absorptance (A) was calculated using the formula A = 1 − R, where T takes a value of 0 because the structure has a 200-nm thick metal layer and a photonic crystal, and it is difficult for visible light to pass through the structure. The measured results are shown in Figure 10. Scanning electron microscopy (SEM) images of the sample were obtained using a Nova NanoSEM 450, and the results are shown in the inset in Figure 10. From the measured reflection spectrum, three reflection dips were observed at wavelengths of 675 nm, 604 nm, and 489 nm, and the reflection rates were 7.7%, 9.9%, and 9.4%, respectively. The three reflection dips were all within the band-gap range of the PC. It can be seen from the measured absorption spectrum that the maximum absorption rates were 92.3%, 90.1%, and 90.6%, respectively. The three structures achieved near perfect absorption at wavelengths of 675 nm, 604 nm, and 489 nm, respectively. According to the relationship between the absorption wavelength and the thickness of the dielectric layer, the thickness of the prepared three structures were offset by about −3.6%, +0.7% and −2.2%, respectively, as compared with the calculated structures. The results showed that the experimental spectra agreed well with the simulated results. The small divergences in these results originated from deviations in the layer thicknesses during the deposition process and from differences between the theoretical and experimental refractive indices of the materials.

## 4. Conclusions

In this study, perfect absorbers were theoretically and experimentally investigated in a photonic heterostructure composed of a truncated all-dielectric photonic crystal (PC) and a thick metal film. Based on the electromagnetic field distributions observed at the absorption wavelengths, it was determined that the perfect absorption characteristics of the structures were dependent on the OTS. The experimental results agreed well with the theoretical values. The absorbers prepared in this study have the advantages of having a simple design and convenient preparation. These perfect absorbers may be applied in the fabrication of new types of optical absorption devices in the future.

## Figures and Tables

**Figure 1 nanomaterials-11-03447-f001:**
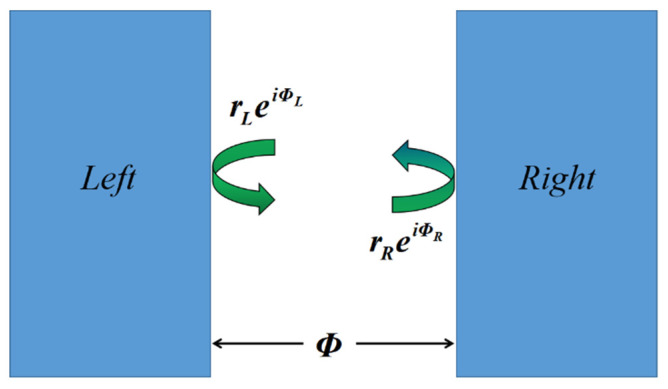
Schematic of the cavity.

**Figure 2 nanomaterials-11-03447-f002:**
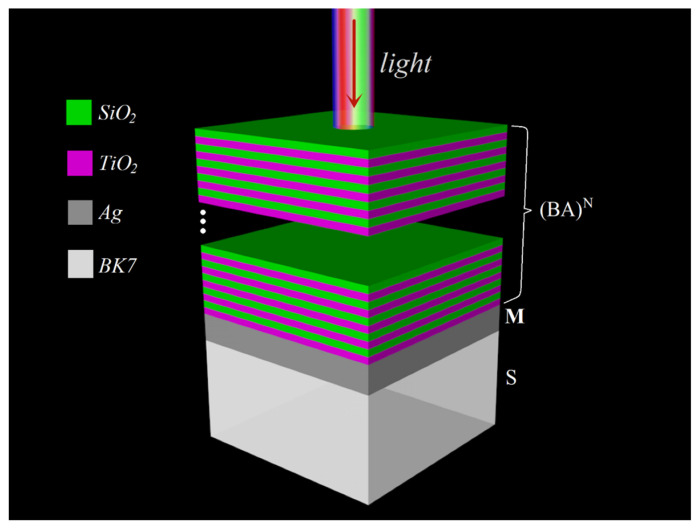
Schematic of the (BA)^N^MS structure. B represents the SiO_2_ layers with n_B_ = 1.431. A represents the TiO_2_ layers with n_A_ = 2.123. M denotes the thick Ag metal layer. S is substrate BK7 with n_S_ = 1.52.

**Figure 3 nanomaterials-11-03447-f003:**
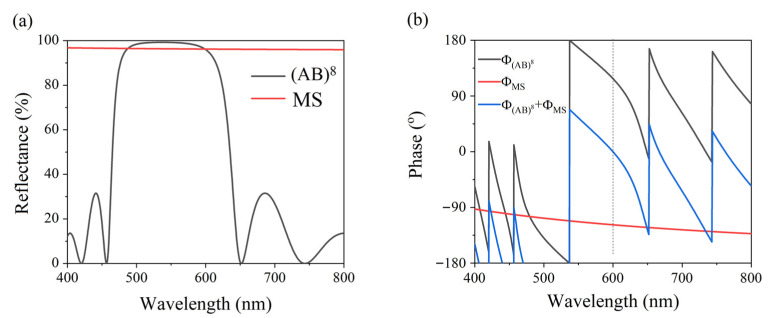
(**a**) Simulated reflection spectra of PC (BA)^8^ (solid black line) and Ag-glass structure MS (solid red line). (**b**) Reflection phases of MS Φ_MS_ (solid black line) and PC Φ_(BA)_^8^ (solid red line) and the sum of Φ_(BA)_^8^ + Φ_MS_ (solid blue line).

**Figure 4 nanomaterials-11-03447-f004:**
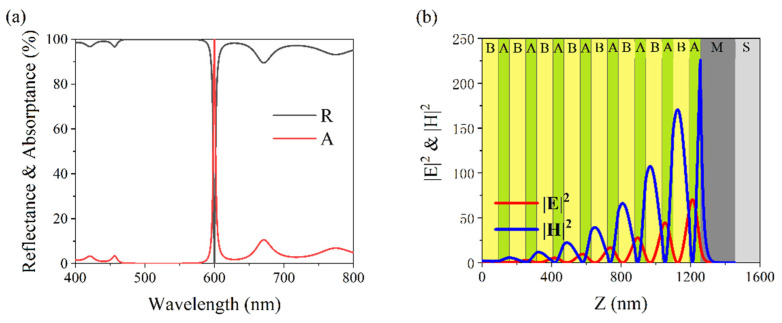
(**a**) Simulated reflection (solid black line) and absorption (solid red line) spectra of (BA)^8^MS. The parameters are: d_A_ = 63.2 nm, d_B_ = 93.8 nm, d_M_ = 200 nm. (**b**) Simulated intensities of |E|^2^ (solid red line) and |H|^2^ (solid blue line) in (BA)_8_MS at an absorption wavelength of 600 nm.

**Figure 5 nanomaterials-11-03447-f005:**
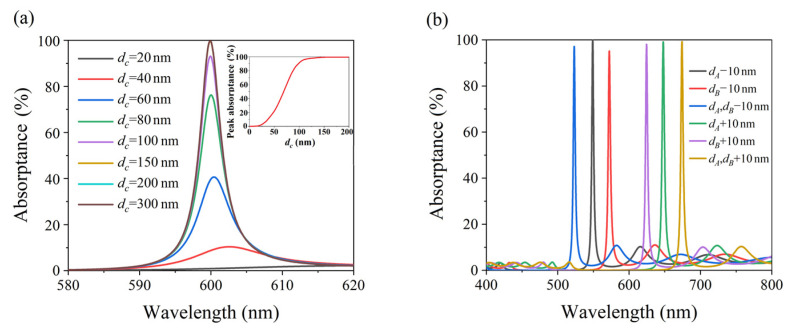
(**a**) Simulated absorption spectra of (BA)^8^MS with different metal layer M thicknesses (d_M_ = 20 nm, 40 nm, 60 nm, 80 nm, 100 nm, 150 nm, 200 nm, 300 nm). The inset figure shows the absorptance of the absorption peak versus the thickness of the metal layer M. (**b**) Simulated absorption spectra of (BA)^8^MS with different dielectric layer thicknesses.

**Figure 6 nanomaterials-11-03447-f006:**
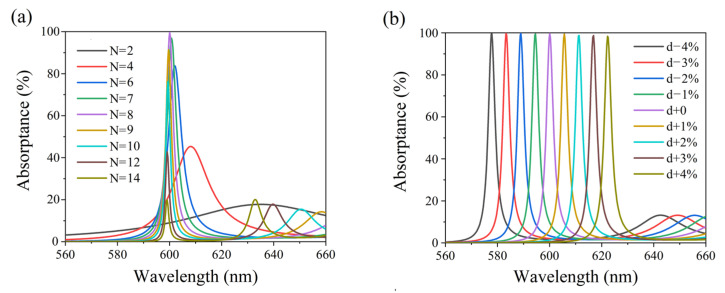
(**a**) Simulated absorption spectra of (BA)^8^MS with different period numbers (N = 2, 4, 6, 7, 8, 9, 10, 12, 14). (**b**) Simulated absorption spectra of (BA)^8^MS with different dielectric layer thicknesses. The thickness of each layer is increased (+1%, +2%, +3%, +4%) or decreased (−1%, −2%, −3%, −4%).

**Figure 7 nanomaterials-11-03447-f007:**
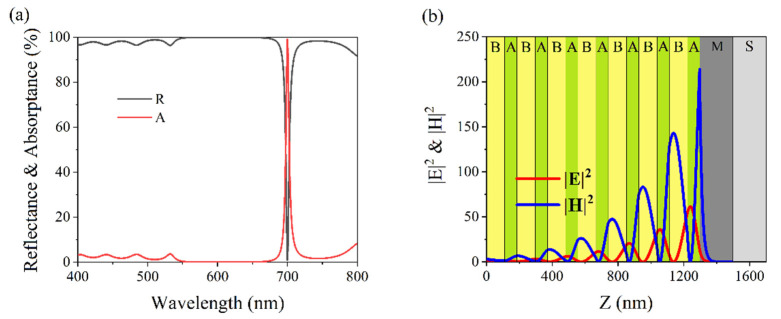
(**a**) Simulated reflection (solid black line) and absorption (solid red line) spectra of (BA)^7^MS. The parameters are d_A_ = 110.7 nm, d_B_ = 74.6 nm, d_M_ = 200 nm. (**b**) Simulated intensities of |E|^2^ (solid red line) and |H|^2^ (solid blue line) in (BA)^7^MS at an absorption wavelength of 700 nm.

**Figure 8 nanomaterials-11-03447-f008:**
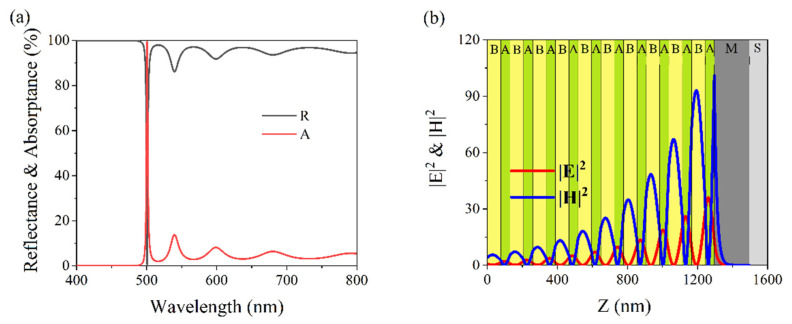
(**a**) Simulated reflection (solid black line) and absorption (solid red line) spectra of (BA)^10^MS. The parameters are d_A_ = 77.4 nm, d_B_ = 52.2 nm, and d_M_ = 200 nm. (**b**) Simulated intensities of |E|^2^ (solid red line) and |H|^2^ (solid blue line) in (BA)^10^MS at an absorption wavelength of 500 nm.

**Figure 9 nanomaterials-11-03447-f009:**
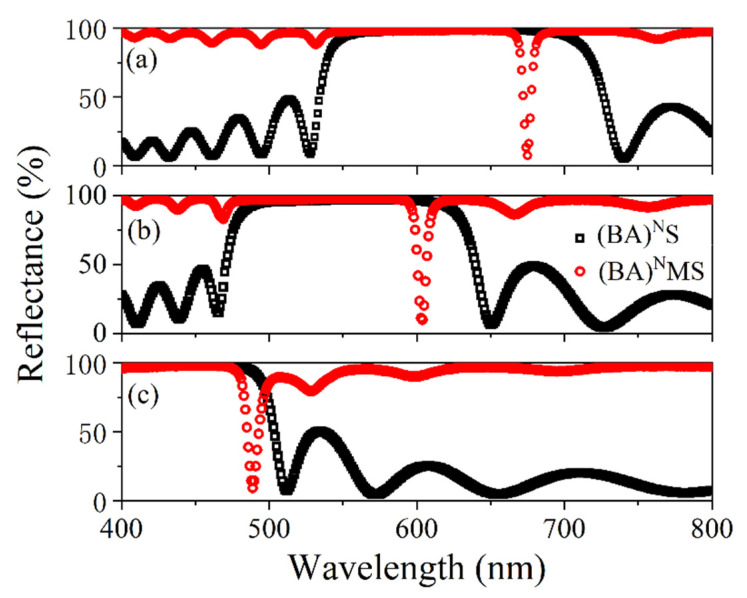
Measured reflection spectra of (BA)^N^MS (red circles) and (BA)^N^S (black circles). (**a**) (BA)^7^MS. (**b**) (BA)^8^MS. (**c**) (BA)^10^MS.

**Figure 10 nanomaterials-11-03447-f010:**
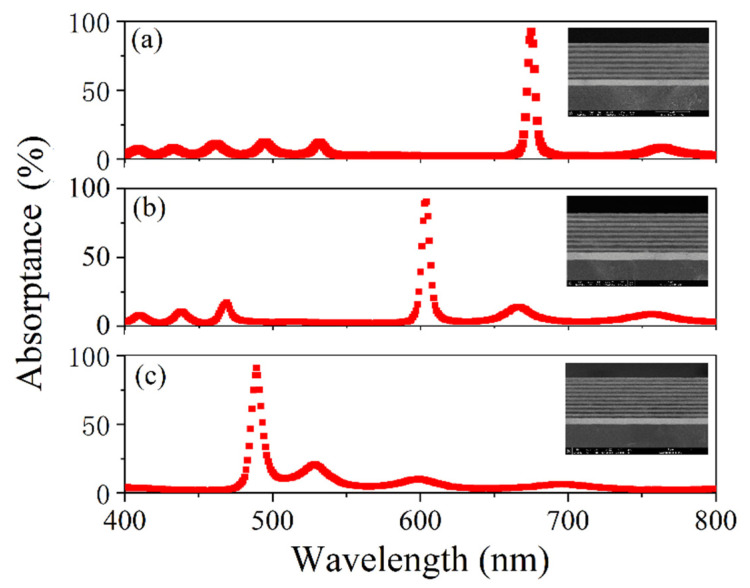
Measured absorption spectra of (BA)^N^MS. SEM images shown in the inset figures. (**a**) (BA)^7^MS. (**b**) (BA)^8^MS. (**c**) (BA)^10^MS.

## Data Availability

The data that support the plots within this paper and other findings of this study are available from the corresponding authors on reasonable request. The transfer matrix simulation code: https://github.com/sduluguang/transferMA.git (14 November 2021).

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
