# Peer review of "Perfect Optical Absorbers by All-Dielectric Photonic Crystal/Metal Heterostructures Due to Optical Tamm State"

_nanomaterials, 2021, doi:10.3390/nano11123447_

Round 1
Reviewer 1 Report
The paper presents a hybrid photonic-crystal—metal structure that has a high degree of optical absorption at a target wavelength. The increased absorption was partially attributed to optical Tamm states. The simulation analysis is solid and it is supported by experimental results. The paper should be suitable for publication after some minor issues are addressed:
- For the transfer matrix analysis, did the author write their own simulation code or did they use a library/solver? Please indicate that in the paper. If the authors developed their own code, then, if possible, please include your code as GitHub repository and cite the repository in the paper. This would allow readers to reproduce the results as well as investigate the phenomenon for themselves. I believe this would greatly increase the impact of the paper.
- Please include a detailed description of the fabrication process. For example, for the thin-film deposition, mention the tool used, chamber pressure, gas flow rate, deposition rate, deposition time etc.
- When using the formula A = 1-R-T, is it assumed that T = 0? Is this true for both the simulation and experimental results? I am assuming that the authors are measuring R in their experiment and finding A by A = 1-R. The same might be true for the simulation analysis. Please confirm in the manuscript.
- In Figure 3, the authors show reference spectra of Ag and the 8-layer AB structure. Do these results consider a glass substrate? Meaning, do the authors perform a simulation of Ag-Glass structure (with air on top) and (AB)8 – Glass? Otherwise, the comparison between Figure 3 and Figure 4 would not be justified. Please clarify.
- If possible, please use a different color scheme to represent SiO2 and TiO2 (perhaps, some lighter shades). Otherwise, the field plots in Figure 4, 6, and 7 are not clear (the plot lines aren’t very visible over the structure colors). The authors may also choose to plot the field lines in a lighter shade to increase contrast.
- Could the authors comment in detail the procedure they used to shift the absorption spectrum from 600nm to 700nm and to 500nm (figure 4, figure 6 and figure 7)? Did they adjust the thickness parameters manually to find the specific absorption wavelength? A trail-and-error based approach? Please mention the optimization approach that was used.
- The authors may cite a few papers related to transfer matrix method for readers who are interested in the details of the approach. For example:
https://doi.org/10.1364/AO.41.003978
https://doi.org/10.1016/j.optlastec.2015.11.011
Reviewer 2 Report
The manuscript reports the design, simulation and experimental validation of “perfect” absorbers at certain wavelengths based on 2D photonic crystals and metallic reflector. The study is interesting, and the authors succeed in creating structures with high absorption at desired wavelengths. The structure of the work is suitable and the results are reported in a logical way. The manuscript has however some points that should be improved:
1) It is recommended that a native speaker proofreads the document. There are some mistakes, and sometimes it is hard to understand some sentences.
2) You mention that solar energy could be an application of this technology. Could you please explain how you envision that the technology could be used?
3) Over the study you are working with 7-8 periods of dielectric layers. Have you studied the effect of changing the period number? Can you please comment on this?
4) The thickness of the dielectric layers is given with sub-nm precision. Could you please provide the errors of these thicknesses? How much precision is required at the manufacturing step in order to obtain a working structure? In other words, the performance of photonic crystals is dependent on the accuracy manufacturing each component. How much performance is lost when the manufacturing steps deviate from the ideal case? This is something that can be simulated.
5) Figure7.b. The X axis should be “Z(nm)”, rather than “Wavelength”.
6) At the abstract: “indicating that the material can be prepared for further applications.” (line 22). Please clarify.
7) At the abstract, when you mention “optical sensor”. This is not a “field”. Please substitute for “optical sensor technology”.
8) Substitute “micro-nano structures” for “micro/nano-structures”.
9) The text is full of redundancies. For example: “We designed the experimental samples and measured the properties of the samples”. Or “composite structure composed of”. Could you please improve the wording?
10) “the light is reflected back and forth in the microcavity, creating a strong local light phenomenon.” Could you please be more accurate with this sentence?
11) “It was the due to the existence (page 5, line 148)”. It doesn’t make sense, please correct it.
12) The data of reference [1] is incorrect.
13) In page 6 you have two paragraphs that are identical, but just changing the parameters values (starting with: “A perfect light absorber was designed for a wavelength of 700nm…”). Could you please rewrite the text so that it is not a perfect copy?
